# Methods for Congenital Thumb Hypoplasia Reconstruction. A Review of the Outcomes for Ten Years of Surgical Treatment

**DOI:** 10.3390/medicina55100610

**Published:** 2019-09-20

**Authors:** Dzintars Ozols, Marisa Maija Butnere, Aigars Petersons

**Affiliations:** 1Department of Hand and Plastic Surgery, Microsurgery Centre of Latvia, Riga East University Hospital, 1000 Riga, Latvia; 2Department of Pediatric Surgery, Riga Stradins University, 1000 Riga, Latviaaigars.petersons@rsu.lv (A.P.); 3Department of Pediatric Surgery, Children Clinical University Hospital, 1000 Riga, Latvia

**Keywords:** thumb hypoplasia, congenital hand deformities, pollicization, toe-to-hand transplantation, tendon transfer

## Abstract

*Background and objectives*: Congenital thumb hypoplasia is a rare deformity of upper extremity. The incidence for thumb hypoplasia grade II–V is 1:10,000 newborns per year in Latvia. A technique for extensor indicis proprius (EIP) tendon transfer with subperiosteal fixation was developed and used for thumb hypoplasia grades II and IIIa. Pollicization or second-toe-to-hand transplantation with metatarsophalangeal (MTP) joint arthrodesis was used for the reconstruction of hypoplasia grade IIIb–V. The aim of this retrospective cohort study is to evaluate the outcomes for reconstruction techniques used in one surgical center during a ten-year period by one surgeon to evaluate functional and aesthetical outcomes for new techniques. *Materials and Methods*: In total, 21 patients were operated on during 2007–2017, and 18 of these patients were involved in this study. Long-term follow-up was completed to evaluate the functions and aesthetics of the hands. Results: disabilities of the arm, shoulder and hand (DASH) was 9.35 (8–10.7) for the second-toe-to-hand with MTP joint arthrodesis transplantation method for pollicization method 19.8 (6–26.7), and for the EIP tendon transposition, 14.54 (0.9–56.3). *Conclusions*: The postoperative functional parameters of congenital hand hypoplasia patients, regardless of the surgical method, are worse than the functional results of healthy patients. The use of the second-toe-to-hand with MTP joint arthrodesis transplantation method provides patients with congenital hand IIIb–V hypoplasia a stable and functional first finger formation. The functional results are comparable to the clinical results of the pollicization method while ensuring the creation of a five-digit hand.

## 1. Introduction

Thumb hypoplasia is a rare congenital deformity that constitutes 3.5% of all upper limb congenital deformities. In Latvia, the incidence of thumb congenital hypoplasia is estimated to be 0.5–2 children per 10,000 newborns, however, precise data on birth defect incidence are not available [1].

The Blauth classification is used to characterize and differentiate the types of thumb hypoplasia [2]. Surgical treatment is required for grade II–V deformities.

For the treatment of grade II–IIIa thumb hypoplasia, it is recommended that the first finger be salvaged by stabilizing the metacarpophalangeal (*MCP*) joint and reconstructing the hypoplastic musculature. The basis for this surgical treatment is to extend the first web space and to stabilize the *MCP* joint [3]. The preferred method is the transposition of the third or fourth finger’s superficial flexor tendon (*FDS*) and fixation through the bone, which establishes stability but reduces grasp strength and causes disarrangement of the fourth and fifth fingers’ flexor system [3]. In 2012, Smith et al. [4] published a report that recommends considering that the pollicization method be used to treat grade IIc as the stabilization of the *MCP* joint is often unsuccessful. In 2007, the Department of Hand and Plastic Surgery, Microsurgery Centre of Latvia, Riga East University Hospital *(MC)*, launched a new reconstruction method—the application of the second finger additional extensor tendon *(EIP)* transposition with subperiosteal fixation.

Pollicization is recommended to create a new finger by using the hand’s second finger and rotating it to the thumb’s location for grade IIIb–V thumb hypoplasia reconstruction [5]. The operation is characterized by well-functioning results, but a four-fingered hand is created. An alternative method can be to use the second-toe-to-hand transplantation technique; however, the method is very complicated and does not provide stable functionality [6]. In 2010, this second-toe transplantation method was developed in the *MC* by utilizing the *MTP* joint arthrodesis in the transplant. *MTP* joint arthrodesis provides a stable and appropriate length of the metacarpal bone for the newly constructed thumb. As the child grows, the carpometacarpal (*CMC)* joint is developed and, therefore, no additional ligament reconstruction is performed [7,8]. The operation and postoperative periods as well as the preliminary functional results are evaluated and justify the effectiveness of this method and the continuation of its application. The functional results of hand reconstructive methods are evaluated by the disabilities of the arm, shoulder, and hand *(DASH)* score, pediatric evaluation of disability inventory *(PEDI)*, range of motion *(ROM)*, and visual analogue scale (*VAS)* international scales. These scales are adapted for interpretation in the international literature [9,10] but the *DASH* score is not optimally tailored to assess the hand functionality results for children of all ages and was therefore used for children older than eight years of age [11]. The aim of the work is to review thumb hypoplasia treatment methods and to evaluate the complex postoperative results in children with congenital thumb hypoplasia for reconstruction techniques used in one surgical center for a ten-year period by one surgeon. Also, the outcomes are evaluated and compared for a new technique for thumb hypoplasia grade IIIb–V reconstruction, which can provide a five-digit hand and restore the functionality of the thumb.

## 2. Materials and Methods

Thumb hypoplasia grade II and IIIa reconstruction uses *EIP* tendon transposition with a subperiosteal fixation operation method. Thumb hypoplasia grade IIIb–V reconstruction uses the new surgical method of second-toe transplantation with *MTP* joint arthrodesis, and the classical pollicization operation method. In the time frame of 2007–2017, 21 patients were examined for congenital thumb hypoplasia, including five bilateral cases. Overall, 25 reconstructive operations were performed for this group of patients at the MC, and 18 of the patients diagnosed with grade II–V thumb hypoplasia were included in the study. The *EIP* tendon transposition method was used for 14 patients, pollicization for two patients, and the remaining two patients had second-toe-to-hand transplantation with *MTP* joint arthrodesis performed.

Analyses of the operation period, postoperative period, and operation complications were performed for patients with grade II–V thumb hypoplasia treated during 2007–2017 via hospitalization and outpatient records of *MC* and Children Clinical University Hospital *(CCUH).* The *DASH* score and *PEDI* questionnaire as well as the determined scores from the *VAS* and *ROM* were used to evaluate the subsequent functional results. For this study, these scores and questionnaires were assessed in person during outpatient visits.

The visual analogue scale *(VAS)* was used to evaluate the patients’ aesthetic data in which the score can range from 1 to 10, with the lower value corresponding to a better evaluation. *VAS* was divided into two components—*v* for visual and *f* for functional. The patients and their parents had to answer the questions for each component: *“Does the reconstructed first finger look like a thumb?”*; *“Does the reconstructed first finger behave like a thumb?”*.

The evaluation of the hand’s functionality consisted of strength/force measurements: grasp and pinch grip. The grip strength is based on the stability of the thumb and the other fingers’ strength, even though the pinch grip is provided by the mobility of the thumb and forefinger. Strength measurements were established for both hands by using a balloon (pneumatic) and a mechanical (janmar) dynamometer. To evaluate the postoperative results for each patient, a comparison was made between the operated hand and the healthy hand. These strength measurements were further compared to the standard functional strength results of children of the corresponding age. These standards were determined by research on the grip strength of 970 patients aged three to seven years conducted by the anthropology laboratory of the Anatomy and Anthropology Institute at Riga Stradins University *(RSU)*.

The data acquired from the retrospective study were systematized in Microsoft Excel 2016 data processing program. The acquired functional data were compared with the population standards respective to that age group (control group). The analysis was realized using IBM *SPSS Statistics v.22 (Statistical Package for the Social Sciences) independent samples t-test (Student’s t-test)*. Data analyses were performed by using *IBM SPSS Statistics v.22 (Statistical Package for the Social Sciences) paired samples t-tests (Student’s t-test)* and Wilcoxon–Mann–Whitney test. Evaluation of statistical hypotheses evaluation used the significance level (*p* ≤ 0.05 for acceptance and *p* > 0.05 for rejection). This study was evaluated by the Ethics Committee of the medical and biomedical research foundation at Riga’s Eastern Clinical University Hospital, and received the authorization no. 16–A/14.06.2014. The parents of the patients participating in this study signed the consent form *(ICF).*

### 2.1. Thumb Hypoplasia Surgical Treatment

Congenital thumb hypoplasia is a wide spectrum of thumb deformities starting, minimally, with a smaller first digit and narrowed web space up to the complete absence of a first digit. The first classification was created by Muller in 1937. The modern classification was developed by Blauth in 1967 and consists of five grades. This classification was updated several times, first by Manske in 1992 involving a division into grade IIIa and IIIb, by Buck-Gramcko in 2002 who added grade IIIc, and finally by Tonkin in 2013 (Figure 1) who added subgrades such as IIc [12,13,14].

#### 2.1.1. Thumb Hypoplasia I–IIIa Reconstruction

Grade I thumb hypoplasia patients have a smaller thumb, slightly narrowed web space, but function normally, therefore, surgical reconstruction is not necessary [15].

##### FDS Oponensplasty (Transposition)

Thumb hypoplasia grade II and IIIa treatment consist, first, of web space reconstruction using several types of flaps, such as Z-plasty or rotational first dorsal metacarpal artery *(FDMA)* flap [16,17] and flexor digitorum superficialis *(FDS)* opponensplasty for stabilization, as described by Kozin and Ezaki [18]. *FDS* from the third or fourth finger are transferred and fixated through the bone of the thumb’s proximal phalanges [4,19,20]. The difference between subtypes IIa, IIb, and IIc are related to the instability of the *MCP* joint. For IIc, instability occurs on several planes and reconstruction is difficult [21]. Recommendations by Smith et al. [4] for grade II hypoplasia treatment include that IIa has to be reconstructed with *FDS* from the fourth finger through bone fixation, and for IIb, *MCP* joint arthrodesis and *Huber* opponensplasty, and for IIc, tendon transfer and *ADM* transfer [4,19]. Mende, Suurmeijer, and Tonkin [21] recommended performing pollicization for grade IIc because of the multiaxial instability of the *MCP* joint.

##### EIP (Extensor Indicis Proprius)

Tendon transfer with subperiosteal fixation was developed in the MC and used since 2007 for the reconstruction of hypoplasia grade II–IIIa. The *EIP* tendon localizes on the ulnar side of the extensor digitorum *(ED)* tendon and is separated through three small incisions: first at the second *MCP* level, second at the wrist level, and third in an S-shape at the level of the first *MCP*. The *EIP* tendon is visualized, detached at the level of second *MCP* joint, and transposed to the thumb. Then, the tendon is pulled through the periosteum and the *MCP* joint capsule, starting from ulnar side to the radial side supporting and strengthening the ligaments. The distal end can be attached to the *EPL* tendon at the *IP* joint level. Fixation is completed by non-absorbable suture 3/0 or 4/0, and then thumb stability can be checked by carefully pulling the *MCP* joint laterally. After a successful test, the *MCP* joint is secured with two K-wires. Cast and wires are removed after five weeks, and then hand therapy has to be started in order to improve the motion of the thumb (Figure 2).

### 2.2. Thumb Hypoplasia Grade IIIb–V Reconstruction

#### 2.2.1. Pollicization 

The pollicization procedure is the method of choice for reconstruction of thumb hypoplasia grades IIIb, IV, and V [22,23]. There are several pollicization or index finger transposition methods, with differences in skin incisions and scar placement (Figure 3). Pollicization methods described by Buck-Gramcko [24], Foucher et al. [25], and Kozin et al. [26] can be found in the literature. 

The index finger is transferred to the position of the thumb in 135° pronation and 45° palmar abduction. The second *MCP* joint is used to create new *CMC* joint. The proximal phalanx of the second digit becomes the metacarpal bone of thumb. Bone fixation can be done using K-wires [24] or intraossal sutures [27]. Index finger tendons provide flexion for the thumb, *ED* and *EIP* provide extension, intrinsic muscles provide abduction and adduction and have to be reattached to extensor mechanisms at the level of the proximal phalanx. Several authors recommend rebalancing of the extensor mechanism. Digital artery or the common digital artery of the index finger provides vasculature. Often, the second finger’s radial digital artery can be hypoplastic or aplastic. Digital nerves provide innervation of the new thumb [28].

#### 2.2.2. Non-Vascular/Vascular Hemi-Metatarsal Bone Transplantation

Chow et al. [29] described the technique using the non-vascularized part of the fourth metatarsal bone. This method can be used to reconstruct thumb hypoplasia grade IIIb and IV. Schneider et al. [30] described a case report for thumb hypoplasia grade IIIb reconstruction using the vascularized second metatarsal bone. Several staged reconstruction operations require reconstructing a stable thumb with a limited range of motion. The benefit for these methods is the possibility to create a five-digit hand without toe count limitations, but donor side morbidity must be considered. Poor functionality of the reconstructed thumb has been reported by Nakada et al. [31]. This method is commonly used in Asian countries.

#### 2.2.3. Vascularized Second-Toe Transplantation

The first microvascular toe-to-hand transplantation was performed in 1964 by Buncke [32] for a rhesus monkey. The first second-toe transfer for thumb reconstruction was performed by Yang [33] in 1966, but the first toe was used for thumb reconstruction by Cobbett [34] in 1968. The first-toe-to-hand transfer for a congenital hand deformity was reported in 1977 by O’Brien [35] when the second-toe transfer for thumb reconstruction was used for two five-year-old girls. Thumb hypoplasia grades IIIb, IV, and V do not have a stable *CMC* joint and, therefore, toe-to-hand transplantation is not recommended [36]. Ozols et al. [37] published the fourth-toe-to-hand transplantation for pediatric patients. The fourth toe is shorter and, therefore, it is difficult maintain length, but donor side morbidity seems to be less substantial than when using the second toe.

#### 2.2.4. Second MTP Joint Transplantation

Foucher et al. [38] described a technique using the second *MTP* joint to reconstruct thumb hypoplasia grade IIIb.

#### 2.2.5. Second-Toe Metatarsal Bone Transfer 

In 2004, Tu et al. [39] described a new technique of second-toe metatarsal bone transfer for thumb hypoplasia grade IIIb, IV, and V reconstruction. This method is based on total second-toe transfer with *MCP* tendon rebalancing and *CMC* joint arthrodesis. Tan et al. [6] showed that pollicization of the index finger remains the gold standard for thumb reconstruction in children with type IV radial deficiency and grades IIIb, IV, or V hypoplastic thumb. The operative time is shorter, the recovery time is quicker, and patients have a better range of motion compared with toe transfer.

#### 2.2.6. Second-Toe-To-Hand with MTP Joint Arthrodesis Transplantation Method 

This method was developed in MC and has been used since 2010 (Figure 4 and Figure 5). The second-toe transplantation with *MTP* joint arthrodesis can be used for congenital absence of the thumb, such as in hypoplasia grade IIIb, IV or V cases, in which parents and patients decided to keep a five-digit hand. *MTP* joint arthrodesis is used to create a stable and long metacarpal bone, and the *PIP* joint becomes the *MCP* joint of the new thumb. Functional outcome results published by the authors showed similar functional results to pollicization functional outcomes, but aesthetical results were evaluated higher as five-digit hand is aesthetical more acceptable [8]. Second-toe-to-hand with *MTP* joint arthrodesis transplantation technique was successfully used to reconstruct grade IIIb thumb hypoplasia for a radial longitudinal deficiency (*RLD)* grade IV (Bayne and Klug classification [40,41]) patient with possible pinch and grasp movements [7].

The use of the second toe with *MTP* joint arthrodesis can be an alternative technique with a promising outcome. Simple *MTP* joint arthrodesis instead of *MTP* joint tendon rebalancing can provide stability and length for a reconstructed metacarpal bone [39]. The average operation time is 4 (4–4.05 h) h, which is much faster than the technique of Tu et al. [39], that requires an average of 8 (6–12 h) h. It seems that careful tendon rebalancing is a time-consuming procedure. A stable pseudo *CMC* joint is much more useful instead of stable fixation as modern touchscreen gadgets require mobility in the thumb and stability is not as important anymore as the number of laborers using hammers and axes are dwindling. The differences of the most common methods used for thumb hypoplasia grade IIIb–V reconstruction are listed in Table 1. 

## 3. Results

Nine children underwent reconstructive surgery during the first year of life (three pollicizations and six *EIP* tendon transpositions). Finger transplants have been performed in children at an average of 86.5 months of age. The average child age during pollicization operations was 13 (9–17) months, while for *EIP* tendon transposition it was 38 (11–128) months. Children had an average age of 86.5 (54–119) months when undergoing a second-toe transplant. In an operational time analysis, it was concluded that the duration of the *EIP* tendon transposition operation is the shortest with an average of 60.71 (30–115) min, the duration of pollicization averaged 92 (50–160) min. The second-toe-to-hand with *MTP* joint arthrodesis transplantation takes an average of 242.5 (240–245) min.

For the *EIP* tendon transposition, the average *DASH* is 14.54 (0.9–56.3). Two patients underwent *EIP* tendon transposition, the *DASH* score is above 50 points, and 30.2 for one patient, indicating poor functional outcome. Two patients with thumb hypoplasia grade IIIa had a reduced metacarpal bone and a stable *CMC* joint, however, one patient had a IIIb-unstable *CMC* joint. In patients who underwent a second-toe-to-hand with *MTP* joint arthrodesis transplantation, the *DASH* averaged 9.35 (8–10.7). In the pollicization group, *DASH* has an average of 19.8 (6–26.7). The *PEDI* questionnaires showed the best results for patients with the second-toe transplant method with 64 points (64–64), the *EIP* tendon transposition patients with 62 points (52–70), while pollicization patients’ functional performance following the questionnaire assessment was 60 points (56–64). Statistical processing of the results of *DASH* and *PEDI* questionnaires resulted in no statistically significant difference (*p* > 0.05) because the number of patients in the comparable groups was too small.

The *VAS(v*) scale rating is measured from points 1 to 10, whereby a lower value corresponds to a better score. The *VAS(v)* data collection shows that weaker results are in the pollicization group of patients. This assessment is related to setting up a four-finger hand in patients following pollicization. An average visual assessment of the thumb of the *EIP* transposition group is 2.85 (1–9) and shows that grade II and IIIa hypoplasia thumbs are only slightly smaller than the normal arm’s thumb, and a good-looking thumb is obtained during finger stabilization. The assessment of the toe-to-hand transplant group is considered to be excellent, although the second toe of the leg is significantly different from a thumb. Perhaps the high assessment is given directly by the creation of a five-finger hand, which seems essential in the eyes of patients and their parents. The score of the *VAS(f)* is measured from points 1 to 10, whereby a lower value corresponds to a better score. The *VAS(f)* data collection shows that the results are similar across all patient groups. The weakest results were observed in patients with grade IIIa hypoplasia. The results of the visual analogue scale (*VAS (v)* and *VAS (f)*) have not produced a statistically significant difference *(p >* 0.05) between the comparable types of surgery because the number of patients in the groups was too small. The analysis of the functional data was performed with nine patients aged three to seven years, and the results compared to the average age norm. There was a statistically significant difference between the results of the patient grip force and the normal variant (*p* = 0.018) using an *independent samples* t-test. The results suggest that operated patients developed finger and hand functionality after thumb reconstruction, which differs significantly from patients without congenital deformation. Congenital hypoplasia of the thumb is not just an isolated deformation of the first finger but the functionality of the other hand, specifically the radial side muscle, and second and third fingers as all playing an essential role in securing the grip of the hand.

Comparing the grasp force measurement data for the healthy and operated arm, 7 out of 18 patients were found to have better results for the healthy arm, and the results of the operated arm were better for only two patients after following the patients with a second-toe-to-hand with *MTP* joint arthrodesis transplantation operation and one patient after pollicization. On the other hand, the four patients in our group had identical measurements of force grip on both hands.

The results of the pinch force measurements for the healthy arm were better for 8 patients out of 18, and only in one case was the force of the observed pinch grip higher than in the healthy arm. Identical measurements of the pinch strength in both hands were observed in four patients. Five patients could not undergo the functional performance assessment due to thumb hypoplasia reconstruction in both hands (Table A2 and Table A3). When comparing the dominant and non-dominant hand, no significant differences in pinch and grasp force results were identified. The healthy hand is superior to the sick hand in any case, regardless of which the child uses as the dominant. The grasp and pinch are provided by multiple combinations of muscles and fingers, while hypoplasia patients are impaired not only by the thumb structures.

## 4. Discussion

Congenital thumb hypoplasia is a rare disease with an incidence of 2:10,000 liveborn newborns a year. In the Republic of Latvia, it is difficult to obtain accurate data on the thumb deformities because there is no unified system for the registry of congenital malformations. While performing a collection of outpatient and hospital data from *CCUH* and *MC*, it was found that the number of congenital thumb hypoplasia patients was, on average, 2–3 patients per year. In addition, combined deformations such as *RLD* (1:50,000) and other complex congenital deformations are diagnosed. Second-toe metatarsal bone transplantation for congenital thumb hypoplasia surgical treatment was described in 2004 when a group of Taiwanese surgeons published a series on 11 treated patients [6,39]. The surgical method is based on the creation of a three-jointed thumb as the *MTP* joint is maintained by the intrinsic musculoskeletal tension, which provides stability. The functional results of the study are considered to be good, although the average time for toe transplantation was 8 h (6.5–12). The application time of the method developed in the *MC* is, on average, 4 h (4–4.05) (Table 1), which is more than twice as fast, and the operation is performed by one surgical team. Asian people, due to cultural and religious differences, choose to preserve a five-finger hand by reconstructing a thumb with a partial unrounded or truncated fragment of metatarsal bone because the loss of a toe to these patients is also unacceptable. 

The method developed by Chow et al. [29] is based on a non-vascularized fourth hemi-metatarsal bone transplant. The study published results for five patients with six operated thumbs. The operation was carried out in two steps: the first phase consisted of an unrounded bone transfer and a second phase of musculoskeletal reconstruction. The published results are judged as fair, although the amount of the thumb movements can only be achieved on the first *CMC* joint and growth in length is reduced. This method is also characterized by several complications, such as graft instability and fracture. The results of Chow et al. [29] for operated patients showed a strength grip on the operated arm of 6.5 kg with a margin of 10.5 kg and a difference of 4.12, or 61.9%, of the healthy arm. Their results for pinch on the operated arm was 1.25 kg and was 3.25 kg for the healthy arm, demonstrating a difference of 2.00 kg, or 38.4% [29]. The results obtained are compared to the second-toe-to hand transplantation with the *MTP* joint arthrodesis method developed by the *MC*—grip force on operated arm of 16 kg, and healthy arm of 18.5 kg, representing a difference of 2.5 kg or 86.4%, but pinch force on operated hand was 0.75 kg, while it was 4 kg for the healthy arm, representing a difference of 3.25 kg or 18.9%. The results of both methods are comparable, but a hemi-metatarsal transplant is supposedly more suitable for Asian people because there is no loss of a toe which is significant for Asian cultural and religious considerations. Functionality is provided by healthy second to fifth fingers, but in *RLD* patients, the second, third, and even fourth fingers often have contractions and movement limitations. Consequently, a second-toe transplant with an *MTP* joint arthrodesis method can offer better functional results [7].

A partial fourth metatarsal bone transplant method for the reconstruction of the thumb hypoplasia grade IIIb has been described in separate cases [30], but the results have not been evaluated. It is stressed that one of the functions of the finger is possible—growth in length, which is significantly smaller than the other hand and has not been evaluated in the donor foot. A partial metacarpal bone transplantation method is applicable for the correction of grade IIIb and IV hypoplasia, but not applicable for the adjustment of grade V (thumb aplasia patients). The method using the second *MTP* joint transplant for reconstruction of thumb hypoplasia IIIb was described by Foucher et al. [38] in 2001, when an overview of cases in three patients was published. The results of the method were described as fair, with the obtained stable thumb providing pinch and grasp [38]. However, the functional results compared to the second-toe-to-hand transplantation with the *MTP* joint arthrodesis method and the method of non-vascularized second metacarpal bone transplantation are to be assessed below—a grasp of 40% of the healthy hand and grip of pinch of 10% of the healthy hand. For the method developed in the *MC*, a grasp of 86.4% and a pinch of 18.9% of the whole healthy hand was observed [8] (Table 2).

The classical surgical method in the reconstruction of thumb hypoplasia grade IIIb–V is pollicization or index finger transposition to the position of the thumb. The good functional results of the method and the relatively short operation time clearly show that it is the first choice for reconstruction in most of the world. The cultural and religious aspects of keeping all fingers, whatever they may be, have contributed to the development of methods of reconstructing with metatarsal bones (vascularized or non-vascularized), although the functional result for the newly formed thumb is weak while all digits of hand and foot are maintained [29]. Tan et al. 2013 study compared the method of second-toe metatarsal bone transplantation with the classical pollicization method, there were significant differences in the operation time—the average duration of surgery in the transplant group was 8 h (6–12 h), but was 2.6 h (2–3.5 h) in the pollicization group. Of all the patients, 80% judged the result after pollicization to be excellent, while in the transplant group, only 60% of the results were assessed as good. The functional results for the range of motion were better in pollicization patients at 74° (60–90°) while transplanted metatarsal patients had 61° (35–85°) [6]. Although the study does not indicate the joint in which the measurements have been performed, it can be concluded that the volume of movements is provided by the stabilized *MTP* joint when assessing the description of the operation and the images included in the article [6,39]. The duration of the second-toe-to-hand with *MTP* joint arthrodesis transplantation method was 4 h (4–4.05), but using the pollicization method, was 1.5 h (50 min–3.5 h). The aesthetic result assessment shows that the appearance of pollicization was lower. Functional results, including the range of motion, of the transplant methods in patients were *IP* joint 27.5° (10–45°), *MCP* joint 70° (50–90°), and *CMC* joint 17.5° (15–20°), while for the pollicization group, *IP* was 51.6° (10–85°), *MCP* was 70° (30–90°), and *CMC* joint was 18.3° (15–20°). The use of second-toe-to-hand with *MTP* joint arthrodesis makes it possible to maintain a functional five-digit hand. The aim of the study was to collect functional and aesthetic results about the applied treatment methods and to perform comparative analyses of the data. The analyses of study data were complicated by the relatively small number of patients (18 patients over 10 years), as well as the various surgical techniques applied that break up the groups of patients. When compiling the functional and aesthetic results for patients in the hypoplasia grade IIIb–V, it was concluded that the functional results did not differ significantly between pollicization and second toe with *MTP* joint arthrodesis transplantation groups, but patients and their parents have a significantly better evaluation of the aesthetic results in the transplant group despite losing a toe. The assessment of *VAS(v)* and *VAS(f)* concluded that the transposition of the *EIP* tendon for the reconstruction of grades IIIa and IIIb of hypoplasia do not produce good results. Possibly, in these patients, and perhaps also grade IIc, one of the methods for total thumb reconstruction should be selected: second-toe-to-hand transplantation with MTP joint arthrodesis or pollicization. 

In evaluating the results of *VAS(v)*, it was concluded that patients in the transposition and transplantation groups had better assessment than patients in the pollicization group. The biggest difference between the groups of these patients is that pollicization patients have four-digit hands, while the transplant and transposition groups have five-digit hands. The Goldfarb et al. [42] 2007 study compared the aesthetic results of pollicization with a thumb, and assessments by the child’s parents, surgeon, and hand therapist. The obtained results showed that none of the created thumbs were assessed as normal. The *VAS* scale was applied, and the results obtained were 6.6 (4.7–9.7) (maximum marking of 10), which are very similar to the results of pollicization operations carried out in the *MC*. The main shortcomings observed in the reconstructed thumbs were that the newly created thumb was too small, short, or long. However, it should be assumed that not only the newly formed thumb [42], but also the total appearance of the whole hand—namely the creation of a hand of four or five fingers—will play an important role in ensuring visual appearance.

The function of both the thumb and other fingers is essential for hand functionality, such as a person’s ability to grasp objects of any size. Historically, the most important function has been to reconstruct the thumb to improve the patient’s work capabilities as the first finger provides up to 50% of the hand’s functionality. At least stable, minimally functional, and preferably sensitive thumbs were required to ensure hand functionality. The motility of the other fingers of the hand—third, fourth, and fifth—while interacting with the thumb, gives the force of grasp and pinch [43]. This historical postulate was long considered the basis for a thumb reconstruction, but the question arises as to whether modern people need a stable, minimally functional thumb that gives the ability to grasp and hold heavy objects, creating an opportunity to work with a shovel, axe, or hammer? Is a less agile thumb more important to modern man, but with possible movements of both flexion and extension, as well as abduction and adduction? Perhaps this less powerful thumb, using touchscreen devices [44], is much more useful for the modern day human than a stable minimally functional thumb. In 2019 Latvian guidelines for the use of reconstructive methods for thumb hypoplasia treatment were developed (Table A1).

## 5. Conclusions

The use of the second-toe-to-hand with *MTP* joint arthrodesis transplantation method provides patients suffering congenital hand IIIb–V hypoplasia with stable and functional first finger formation, which provides good functional results and ensures the creation of a five-digit hand. The operation time of the second-toe-to-hand with *MTP* joint arthrodesis transplantation method is longer than that of the pollicization, and the patient hospitalization is also longer than the pollicization method, but it provides the creation of a five-digit functional hand, providing a better aesthetic outcome. The achieved aesthetic look of the five-digit hand created by this new surgical method is better than the acquired aesthetic look of the four-digit hand of the classical pollicization method. The reconstruction of congenital thumb hypoplasia grade II and IIIa, using the *EIP* tendon transposition method, ensures the establishment of a functional thumb. The postoperative functional parameters of congenital hand hypoplasia patients, regardless of the surgical method, were worse than the functional results of the hand of healthy patients.

## 6. Patents

P-18-103 10.12.2018. The Combined Silicon and Polyurethane Artificial Model for the First Finger Joints Movement Evaluation before Second-Toe-to-Hand Transplantation.

P-18-104 10.12.2018. The External Method for Dynamical Effectiveness Evaluation of Tissue Function in the Metacarpophalangeal Joint Contractures after Long-Term Thumb Reconstruction for Pediatric Patients.

## Figures and Tables

**Figure 1 medicina-55-00610-f001:**
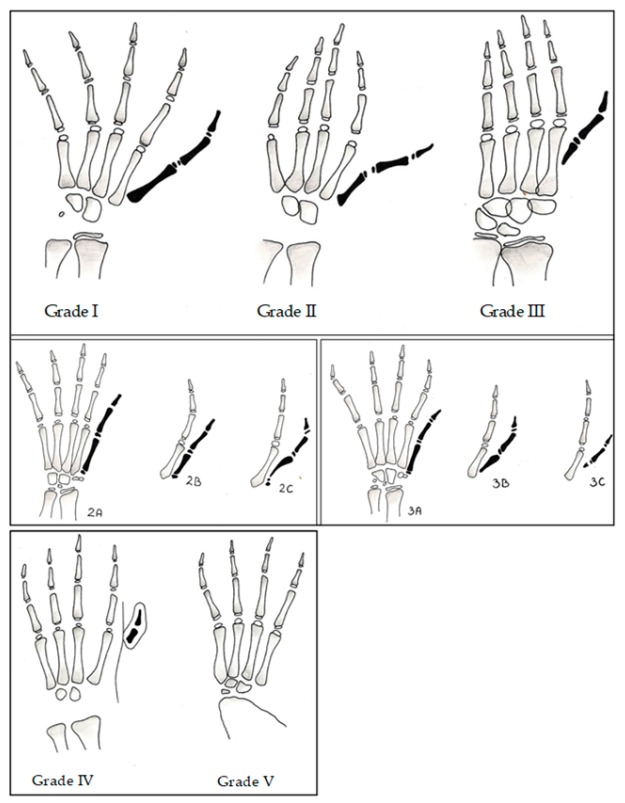
Thumb hypoplasia classification Tonkin et al. 2013 [2].

**Figure 2 medicina-55-00610-f002:**
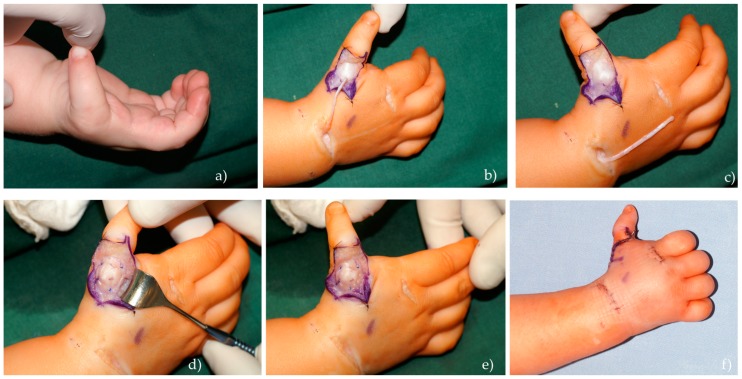
Extensor indicis proprius (*EIP*) transfer with subperiosteal fixation. (**a**) Grade II hypoplasia patient with instability of the metacarpophalangeal MCP joint; (**b**,**c**) *EIP* tendon transfer through three incisions; (**d**,**e**) subperiosteal fixation of transferred tendon and stabilization test; (**f**) final fixation with K-wires.

**Figure 3 medicina-55-00610-f003:**
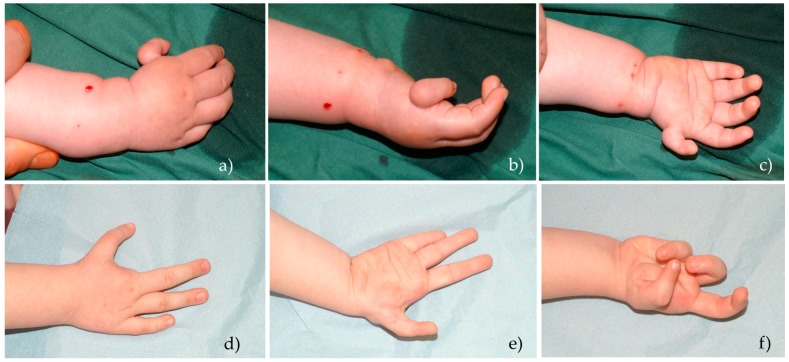
Index finger pollicization procedure [25]. (**a**–**c**) Grade IV thumb hypoplasia before operation; (**d**–**f**) Aesthetical and functional results at an 8-year follow-up visit.

**Figure 4 medicina-55-00610-f004:**
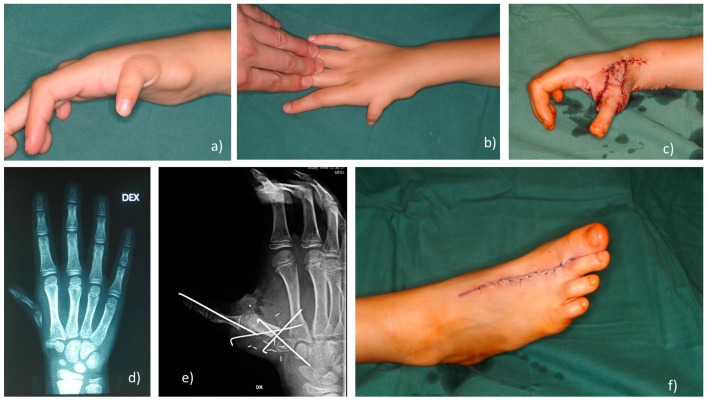
The second-toe-to-hand with *MTP* joint arthrodesis transplantation method. (**a**,**b**) Grade IIIb thumb hypoplasia; (**c**) Toe-to-hand transplantation; (**d**) X-ray for grade IIIb hypoplasia (subluxation of *MCP* joint aplasia of proximal part of metacarpal bone); (**e**) k-wires fixation, *MTP* joint arthrodesis; (**f**) Donor side.

**Figure 5 medicina-55-00610-f005:**
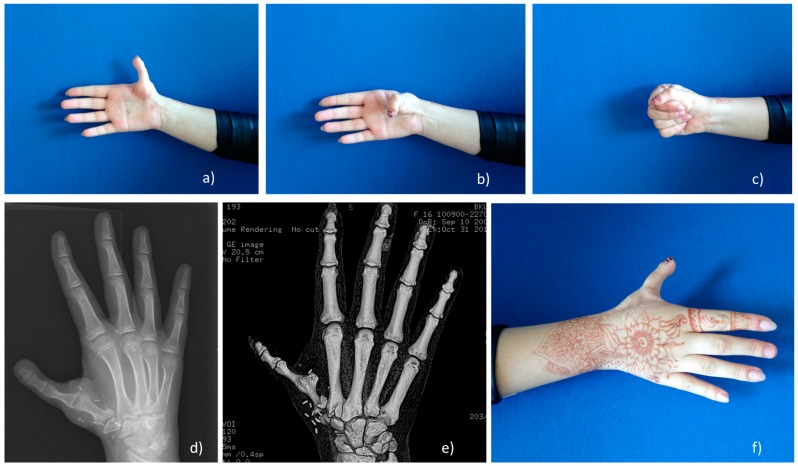
The second-toe-to-hand with *MTP* joint arthrodesis transplantation method outcome. (**a**–**c**,**f**) functional and aesthetical outcome at 6-year follow-up; (**d**) X-ray with transplanted toe, neo*CMC* joint develops; (**e**) 3D CT view of stable, long 48 mm metacarpal bone (equal to contralateral side).

**Table 1 medicina-55-00610-t001:** Differences in the main reconstructive techniques.

Methods	Pollicization	Second-Toe Metatarsal Bone Technique	Second-Toe-To-Hand with *MTP* Joint Arthrodesis Technique
Number of phalanges	Two	Three	Two
Number of joints	Three joints: neoCMC; PIP as MCP; DIP as IP	Four joints: stable CMC, MTP; PIP; DIP	Three joints: pseudo CMC; PIP as MCP; DIP as IP
Operation time	1.5 h (0.50–3.5 h)	8 h (6–12 h)	4 h (4–4.05 h)
Number of fingers in hand	Four	Five	Five

**Table 2 medicina-55-00610-t002:** Functional results for the main reconstructive (five-digit hand) methods.

Functionality Percentage from Healthy Hand	Second-Toe with *MTP* Joint Transplantation Method Ozols et al. [8]	Vascularized Second *MTP* Joint Transplantation Method Guy et al. [38]	Non-Vascularized Hemi-Metatarsal Bone Transfer Chow et al. [29]
Grasp (%)	86.4	40	61.9
Pinch (%)	18.9	10	38.4

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
