# Peer review of "Methods for Congenital Thumb Hypoplasia Reconstruction. A Review of the Outcomes for Ten Years of Surgical Treatment"

_medicina, 2019, doi:10.3390/medicina55100610_

Round 1
Reviewer 1 Report
I think that the paper is of interst and worth to be published; I have only 2 points:
The introduction part is too Long; it should be reduced by 30%. Also other parts seem to be too Long. Please check and reduce.
Some of the Facts in the "Materials and Methods" Part should be placed in the "Introduction" Part.
Please state and discuss the limitations of this study! Please mention the real new data and discuss them!
Author Response
Elements of Revision by: Dz.Ozols, M.M.Butnere, A.Petersons
Manuscript: Methods for the congenital thumb hypoplasia reconstruction. A review of the outcomes for ten years of surgical treatment.
Dear editor,
We would like to thank for the valuable suggestions upon our submitted manuscript. Each of these suggestions was carefully considered and thus the manuscript has been modified accordingly. We are firmly convinced that the careful revision of our work regarding to your suggestions has significantly improved the quality of our paper. Once again, thank You for the motivating review.
Response to reviewers
The reviewers’ suggestions have been addressed point by point as follows and major changes are remarked using track changes in revised version of manuscript:
Reviewer #1:
The introduction part is too Long; it should be reduced by 30%. Also other parts seem to be too Long. Please check and reduce.
Manuscript has been reduced.
Some of the Facts in the "Materials and Methods" Part should be placed in the "Introduction" Part.
Done. Parts “Introduction and materials and methods” were shorten.
Please state and discuss the limitations of this study! Please mention the real new data and discuss them!
Done. Limitations of this study were included in discussion.

Reviewer 2 Report
Review Medicina
Methods for the congenital thumb hypoplasia reconstruction. A review of the outcomes for ten years of surgical treatment. Authors: Ozols, D, Butnere MM, Petersons A.
Format. This manuscript is very massive (20 densely written pages, 7 figures and 4 tables) and I am unsure whether the format is correct for Medicina or not. It is not clearly described whether this is a review article or a retrospective cohort study of the 18 operated patients. If a review on this topic was not specifically requested by the journal, I suggest that the manuscript is rewritten strictly as a retrospective cohort study. In the present version it is very difficult to distinguish the results for the treated patients from general comments and reports from the literature.
A large cohort of 970 children aged 3-7 years examined at an anatomical Institution are mentioned in the text as a reference population, but I cannot find any further information on this study or whether that data has been published?
Also at page 3 line 113-117 an “evaluation study on aesthetic results” with 285 respondents 19-30 years of age is mentioned. It is difficult to understand how this data fits into the retrospective cohort study? When and how was this study performed, has it been published? How were these respondents selected? 49% were healthcare professionals so this could hardly be defined as a normal population. Which questions were asked, were the respondents shown pictures of hands?? On page 10 lines 378ff, I can find some information, but methodological data is lacking.
In material and methods, the ages of the operated patients are not reported and I fail to find that information anywhere in the manuscript. On page 164 it is stated that the inclusion criterium was age 0-18, but the specific ages of the operated patients seem not be reported?
Follow-up time is also not reported (or at least very difficult to find in the manuscript). Of course, it is crucial to follow all growing individuals until the end of growth.
In total, this study included 18 operated patients; 14 reconstructions with the EIP tendon transposition method, 2 pollicisations and 2 toe transfers. These groups are not at all comparable preoperatively and the material is far too small to allow for any statistical comparisons between methods. Patient reported outcomes (such as DASH or VAS evaluations) are furthermore not easily compared between groups in small size retrospective studies, because of large individual variations. To know whether the operations have led to improved patient reported outcome it is advised to do a prospective study comparing pre- and postoperative responses for each individual separately.
The EIP transposition method. As I understand it the tendon is not used as an opponensplasty but pulled dorsally to stabilize the MP joint? What about opposition, how was that achieved with this method?
Toe transfer with MTP joint fusion for hypoplasia grade IIIb-V is suggested, but both two cases operated were type IIIb. How could stability of the missing CMC 1 joint be achieved in a type IV or V hypoplasia? Since follow-up times are not reported the long-term results cannot be evaluated.
Methods. The instrument DASH is not validated for use below 18 years of age and is not relevant in children since the activities in the questionnaire are usually not performed by children. The PEDI has been developed mainly for children with general functional disabilities, such as cerebral palsy and autism. Furthermore, it is intended for children between 6months and 7,5 years and the questionnaire is to be completed by parents or caregivers, not the patients themselves. Regarding perceived function or esthetics it is best to ask the patients themselves and not the caregivers. The PEDI consists of different domains, the relevant one for upper extremity function is the daily activities domain, the others are less relevant and are probably normal in this group of patients. It is unclear which domains that have been used in this study. To my experience the PEDI is not very useful for evaluating results after hand surgery as it does not measure the most relevant aspects. A functional test, such as the Jebsen Taylor, the Box-and Block test or Sollerman test (if adult patients) would have been more appropriate.
In conclusion,
This study reports on a 10 year experience of operating hypoplastic thumbs in Latvia and I want to give credit to the main author for the clinical work and the good results in the 18 patients.
However, the study unfortunately lacks both in study design, instruments for evaluation and the use of statistics. Another problem making it difficult to evaluate the results of the operated patients is the format and structure of the manuscript which lacks a lot of important information about the examined patients. A suggestion could be to rewrite this as a short retrospective cohort study on the 18 patients.
Author Response
Elements of Revision by: Dz.Ozols, M.M.Butnere, A.Petersons
Manuscript: Methods for the congenital thumb hypoplasia reconstruction. A review of the outcomes for ten years of surgical treatment.
Dear editor,
We would like to thank for the valuable suggestions upon our submitted manuscript. Each of these suggestions was carefully considered and thus the manuscript has been modified accordingly. We are firmly convinced that the careful revision of our work regarding to your suggestions has significantly improved the quality of our paper. Once again, thank You for the motivating review.
Format. This manuscript is very massive (20 densely written pages, 7 figures and 4 tables) and I am unsure whether the format is correct for Medicina or not. It is not clearly described whether this is a review article or a retrospective cohort study of the 18 operated patients. If a review on this topic was not specifically requested by the journal, I suggest that the manuscript is rewritten strictly as a retrospective cohort study. In the present version it is very difficult to distinguish the results for the treated patients from general comments and reports from the literature.
Done. Manuscript was reduced to cohort study reporting outcome for congenital thumb hypoplasia treatment 18 cases in 10 year period.
A large cohort of 970 children aged 3-7 years examined at an anatomical Institution are mentioned in the text as a reference population, but I cannot find any further information on this study or whether that data has been published?
Result for anatomical institutoions is not yet published. Data were collected to established fizical evaluation in children population in Latvia. Grasp force date is just one parameter from that study.
Also at page 3 line 113-117 an “evaluation study on aesthetic results” with 285 respondents 19-30 years of age is mentioned. It is difficult to understand how this data fits into the retrospective cohort study? When and how was this study performed, has it been published? How were these respondents selected? 49% were healthcare professionals so this could hardly be defined as a normal population. Which questions were asked, were the respondents shown pictures of hands?? On page 10 lines 378ff, I can find some information, but methodological data is lacking.
Aesthetical study to compare pollicization and toe-to hand transplantation results was done in 2018 and results are published: Ozols, Dzintars, Jānis Zariņš, and Aigars Pētersons. "Long-Term Evaluation of The Functional and Esthetical Outcomes for The New Method of The Toe-To-Hand Transfer for Full-Length Thumb Reconstruction in Congenital Thumb’s Hypoplasia in Children." In Proceedings of the Latvian Academy of Sciences. Section B. Natural, Exact, and Applied Sciences., vol. 73, no. 2, pp. 171-176. Sciendo, 2019.
In material and methods, the ages of the operated patients are not reported and I fail to find that information anywhere in the manuscript. On page 164 it is stated that the inclusion criterium was age 0-18, but the specific ages of the operated patients seem not be reported?
Inclusion and exclusion criteria was removed from manuscript. Tetxt was rewriten: 21 patients were examined for congenital thumb hypoplasia including five bilateral cases. 25 reconstructive operations were performed for this group of patients at the Department of Hand and Plastic Surgery, Microsurgery Centre of Latvia, Riga East University Hospital (MC) and 18 patients diagnosed with grade II – V thumb hypoplasia included in study. The EIP tendon transposition method was used for 14 patients, pollicization for two patients, and the remaining two patients had second toe transplantation with MTP joint arthrodesis performed.
Follow-up time is also not reported (or at least very difficult to find in the manuscript). Of course, it is crucial to follow all growing individuals until the end of growth.
Table A3 included with follow up data for all patients. All patients is still followed and results will be updated in next ten years.
In total, this study included 18 operated patients; 14 reconstructions with the EIP tendon transposition method, 2 pollicisations and 2 toe transfers. These groups are not at all comparable preoperatively and the material is far too small to allow for any statistical comparisons between methods. Patient reported outcomes (such as DASH or VAS evaluations) are furthermore not easily compared between groups in small size retrospective studies, because of large individual variations. To know whether the operations have led to improved patient reported outcome it is advised to do a prospective study comparing pre- and postoperative responses for each individual separately.
Agree. Groups are small and outcome date is difficult to compare. All individual date is listed in appendix table 2 and 3.
The EIP transposition method. As I understand it the tendon is not used as an opponensplasty but pulled dorsally to stabilize the MP joint? What about opposition, how was that achieved with this method?
EIP tendon is attached to MCPJ radial side and this radial side attachment works like abductor.
Toe transfer with MTP joint fusion for hypoplasia grade IIIb-V is suggested, but both two cases operated were type IIIb. How could stability of the missing CMC 1 joint be achieved in a type IV or V hypoplasia? Since follow-up times are not reported the long-term results cannot be evaluated.
MTP transfer follow up time for patient at figure 7 is six years (time for clinical examination). Surgery was done in 2010 next follow up is planned after 10 years post op. There is now complains about stability in CMC joint. All patient operated using this method developed stable neoCMC joint. Second patient outcome was reported in Journal of hand surgery: Ozols, Dzintars, Janis Zarins, and Aigars Petersons. "Bilateral second toe transfers to reconstruct radial longitudinal deficiency: A case report." Journal of Hand Surgery (European Volume) 43, no. 10 (2018): 1114-1116.
Methods. The instrument DASH is not validated for use below 18 years of age and is not relevant in children since the activities in the questionnaire are usually not performed by children. The PEDI has been developed mainly for children with general functional disabilities, such as cerebral palsy and autism. Furthermore, it is intended for children between 6months and 7,5 years and the questionnaire is to be completed by parents or caregivers, not the patients themselves. Regarding perceived function or esthetics it is best to ask the patients themselves and not the caregivers. The PEDI consists of different domains, the relevant one for upper extremity function is the daily activities domain, the others are less relevant and are probably normal in this group of patients. It is unclear which domains that have been used in this study. To my experience the PEDI is not very useful for evaluating results after hand surgery as it does not measure the most relevant aspects. A functional test, such as the Jebsen Taylor, the Box-and Block test or Sollerman test (if adult patients) would have been more appropriate.
I agree that Jebsen test is useful to evaluate hand functionality and DASH and PEDI scores are not the best option to evaluate and compare outcome both scales were usedas there is difference in ages in population groups. DASH score is used for pediatric patients there are articles validating DASH score for pediatric population starting age 8 years: Quatman-Yates, Catherine C., Resmi Gupta, Mark V. Paterno, Laura C. Schmitt, Carmen E. Quatman, and Richard F. Ittenbach. "Internal consistency and validity of the QuickDASH instrument for upper extremity injuries in older children." Journal of Pediatric Orthopaedics 33, no. 8 (2013): 838-842.
To evaluate functionality Grasp and Pinch forces were measured for both hands and data were compared to healthy hand and to Latvian population date.
This study reports on a 10 year experience of operating hypoplastic thumbs in Latvia and I want to give credit to the main author for the clinical work and the good results in the 18 patients.
Thank you very much for evaluating results.
However, the study unfortunately lacks both in study design, instruments for evaluation and the use of statistics. Another problem making it difficult to evaluate the results of the operated patients is the format and structure of the manuscript which lacks a lot of important information about the examined patients. A suggestion could be to rewrite this as a short retrospective cohort study on the 18 patients.
Done. Manuscript rewrite as cohort study to evaluate outcome for operated patients.
